# Validation of UHPLC-ESI-MS/MS Method for Determining Steviol Glycoside and Its Derivatives in Foods and Beverages

**DOI:** 10.3390/foods12213941

**Published:** 2023-10-27

**Authors:** Yollada Phungsiangdee, Pimpuk Chaothong, Weeraya Karnpanit, Pharrunrat Tanaviyutpakdee

**Affiliations:** 1Master of Science Program in Toxicology and Nutrition for Food Safety, Institute of Nutrition, Mahidol University, Nakhon Pathom 73170, Thailand; yollada.phung@gmail.com; 2Food Toxicology Unit, Institute of Nutrition, Mahidol University, Nakhon Pathom 73170, Thailand; pimpuk.cha@mahidol.edu; 3School of Science, Western Sydney University, Locked Bag 1797, Penrith, NSW 2751, Australia; w.karnpanit@westernsydney.edu.au

**Keywords:** steviol glycoside, UHPLC-ESI-MS/MS, validation of method, foods and beverages

## Abstract

The aim of this study was to validate a method for determining nine types of steviol glycoside and its derivatives in food and beverage products, using ultrahigh-performance liquid chromatography tandem mass spectrometry with electrospray ionization (UHPLC ESI MS/MS). The performance characteristics of the analysis method were determined along with their suitability for the intended use. Coefficient of determination (R^2^) calibration curves from 0.2 to 1.0 mg L^−1^ were in the ranges of 0.9911–0.9990, 0.9939–1.0000 and 0.9973–0.9999 for a beverage, yogurt and snack, respectively. Intra-day precisions in terms of percent relative standard deviation (% RSD) of concentration, at 0.2, 0.5 and 1.0 mg L^−1^, for the beverage, yogurt and snack were lower than 15% (1.1–9.3%). At all concentrations, percentage recoveries were in the accepted range of 70–120%. For the matrix effect study, matrix-matched calibration was used for all compounds, obtaining a linear concentration range from 0.2 mg L^−1^ to 1.0 mg L^−1^. Almost all matrix-matched results presented as percentage recoveries were within the accepted range of 80–120%. The limit of detection (LOD) for steviol glycosides ranged from 0.003 to 0.078 μg g^−1^, while the limit of quantitation (LOQ) ranged from 0.011 to 0.261 μg g^−1^. These results indicate that the modified test method can be applied to determine the presence of steviol glycoside and its derivatives in a wide range of sample matrices.

## 1. Introduction

The World Health Organization recommends reducing the consumption of free sugar from daily consumed foods to less than 10% of total energy needs, or to less than 5% for additional health benefits, especially for the reduction of non-communicable diseases (NCDs) [1]. The consumption of free sugar and sugar-containing processed foods and beverages is specified as a cause of NCDs [1]. There are several campaigns used for the reduction of free sugar consumption, including the sugar tax measure. In Thailand, the sugar tax measure started in September 2017. Accordingly, the use of various kinds of artificial sweeteners has increased, including steviol glycoside and its derivatives. 

Steviol glycoside and its derivatives are compounds containing *Stevia rebaudiana* Bertoni. The main steviol glycosides are stevioside, rebaudioside A, rebaudioside C and dulcoside A. The most abundant compounds are rebaudioside A and stevioside (Figure 1) [2,3,4]. Stevioside is 250–300 times sweeter than sucrose and has a bitter aftertaste, while rebaudioside A is 350–450 times sweeter than sucrose and does not have an aftertaste [5]. 

These compounds have been widely used as natural no-calorie sweeteners and sugar substitutes in the food and beverage industry [6]. The use of steviol glycoside and its derivatives is approved in several countries such as Brazil, Argentina, Paraguay, China, Korea and Japan [7]. In 2011, steviol glycosides were approved for use as food additives by the European Food Safety Authority. The Joint Expert Committee on Food Additives (JECFA) and the European Food Safety Authority (EFSA) assigned an accepted daily intake (ADI) of 4 mg steviol equivalents per kg body weight per day, which was brought into practice in December 2011, and a new version of the specifications for steviol glycosides was developed in 2017 [8,9,10]. In Thailand, steviol glycosides have been approved for use as food additives according to the Notification of the Ministry of Public Health No. 418 B.E. 2563 (2020) [11]. Steviol glycosides are allowed to be used in 14 food categories (53 sub-categories), with the limit of the range of allowance in the maximum use set to 30 to 2500 mg kg^−1^. Although safety evaluations have been conducted by multiple regulatory bodies, there is currently no consensus on the toxicological properties of Stevia extracts. Thus, for the protection of consumer health, the risk assessment principle is used to estimate the risk of exposure to steviol glycoside and its derivatives. An important step in risk assessment is exposure assessment, which is calculated based on the consumption amount, concentration of the chemicals and average body weight in the population. There are many techniques used to determine the concentration of steviol glycoside and its derivatives [12], including high-performance thin-layer chromatography (HPTLC) [13], enzymatic hydrolysis [14,15], capillary electrophoresis [16], near-infrared spectroscopy, ion-exchange resin chromatography [17,18], high-performance liquid chromatography (HPLC) with UV detection [19,20,21,22], LC/MS [23] and UHPLC–MS [2]. Nowadays, mass spectrometry is a powerful analytical technique based on ion separation; therefore, ionization is of importance for high sensitivity and selectivity. The modern technique HPLC–MS/MS is the current method of choice for the analysis and confirmation of the intense sweetener concentration in foods [24]. These mentioned methods vary in their performance, such as in terms of sensitivity, precision and accuracy. According to the recommendation of FAO/WHO JECFA, the most appropriate method for determining the steviol glycoside content is HPLC [25,26,27]. A method used for the determination of the concentration of the chemical of interest must be evaluated for its characteristics, which is termed method validation. The requirement of method validation is to confirm that the method can produce accurate and precise results within the scope of analysis. There are many guidelines related to method validation such as AOAC [28], EMA [29], EURACHEM [30], FDA [31], ICH [32], IUPAC [33], NordVal [34], SANCO [35] and the ASEAN Guideline on Analytical Validation [36]. Generally, the validation of an analysis method has been performed by investigating the following performance characteristics: linearity of signal, matrix effect, specificity, precision, accuracy, limit of detection and limit of quantitation, ruggedness and robustness [30,37,38,39].

The data from method validation, especially the limit of detection (LOD) and the limit of quantitation (LOQ), are common parameters used to assess the sensitivity of analytical methods and for dealing with “not detected” analysis results. These parameters are also required to deal with left-censored data, i.e., data below an LOD/LOQ for which the true value is unknown [38,40]. As mentioned above, the concentration of the chemical of interest is an important piece of data in the exposure calculation. In risk assessment, to protect consumer health, LOD or LOQ values are taken into account in cases where the analysis value is “not detected”. This study presents the parameters used in this method validation and confirms that the analytical method used, ultrahigh-performance liquid chromatography tandem mass spectrometry with electrospray ionization (UHPLC-ESI-MS/MS), can accurately and precisely detect steviol glycosides. Furthermore, given that there are environmental concerns about the growing use of steviol glycosides, and given that there is a lack of robust environmental monitoring [41], in this study, low concentrations were used that may be applied to develop and validate an innovative and ecological analytical method. 

## 2. Materials and Methods

### 2.1. Chemicals, Reagents and Standards

Acetonitrile (CH_3_CN) of HPLC grade was purchased from J.T.Baker (Radnor, PA, USA). Ultrapure water (18.2 MΩ/cm) was produced using the Milli–Q IQ 7000 from Merck (Darmstadt, Germany). Formic acid (98–100% purity) was purchased from Merck (Darmstadt, Germany). Steviol glycoside and its derivatives, rebaudiosides A (99.5% purity), rebaudiosides B (99.1% purity), rebaudiosides C (99.3% purity), rebaudiosides D (96.8% purity), rebaudiosides F (100% purity), rubusoside (99% purity), dulcoside A (98% purity) and stevioside (98.4% purity), were purchased from Sigma (St. Louis, MO, USA). Steviolbioside (87% purity) was purchased from Phytolab (Vestenbergsgreuth, Germany).

### 2.2. Standard Solution of Steviol Glycoside and Its Derivatives

A stock mixture of steviol glycoside and its derivatives was prepared at the final concentration of 10 mg L^−1^ by weighing 0.001 g of each standard and dissolving them with deionized water/acetonitrile (80:20 *v*/*v*) in a 100 mL volumetric flask. The stock solutions were stored at −20 °C. 

### 2.3. Sample Preparation and Extraction

There were three types of test matrices (sample blank) for the method validation: non-alcoholic beverage, yogurt and snack. All samples free of steviol glycoside were purchased from retail markets. The ingredients of each blank sample are listed in a Appendix A. The net volume of the non-alcoholic beverage was 500 mL per pack. The three packs of the non-alcoholic beverage were poured into the beaker and mixed well with a magnetic stirrer. For yogurt, the net weight of each pack was 110 g. The three packs of yogurt were transferred into a blender and homogenized until well mixed. For the snack, the net weight of each pack was 36 g. Three packs of the snack were frozen and then homogenized into fine granules. The pooled sample of each type was stored at −20 °C. Samples were analyzed within one month.

### 2.4. Sample Preparation for the Analysis of Steviol Glycoside and Its Derivatives

The analysis of steviol glycoside and its derivatives was modified from that of Gardana and Simonetti [42]. In brief, for the non-alcoholic beverage, one gram of homogenized sample was weighed and then all samples were dissolved in 5 mL of deionized water/acetonitrile (80:20 *v*/*v*), mixed using vortex for 1 min and sonicated in an ultrasonic bath for 10 min. The volume of the solution was adjusted to 10 mL with deionized water/acetonitrile (80:20 *v*/*v*) and filtered through Whatman filter paper number 1. 

For the yogurt, two grams of the homogenized sample were weighed and added to 1.5 mL deionized water and then incubated in a 60 ^o^C water bath for 10 min. The mixture was left to cool at RT before 2 mL each of Carrez I and Carrez II and 1 mL of acetonitrile were added. The mixture was diluted with deionized water to 10 mL before centrifugation at 4000 rpm for 1 h. The supernatant was filtered through Whatman filter paper number 1 and through SPE (500 mg). A C18 cartridge (SPE) was conditioned with 5 mL of MeOH and 5 mL of deionized water. Next, 3 mL of supernatant was loaded and eluted with 3 mL of MeOH, followed by evaporation until dried. The residue was re-dissolved with 3 mL of deionized water. 

For the snack, one gram of homogenized sample was weighed and dissolved in 5 mL of deionized water/acetonitrile 80:20 *v*/*v*, mixed with vortexing for 1 min and sonicated in an ultrasonic bath for 15 min. Next, 1 mL each of Carrez I and Carrez II was added and then the mixture was diluted with deionized water to 10 mL before centrifugation at 4000 rpm for 1 h. The supernatant was filtered through Whatman filter paper number 1 and through SPE (500 mg). A C18 cartridge was conditioned with 5 mL of MeOH and 5 mL of deionized water, loaded with 3 mL of supernatant, eluted with 3 mL of MeOH and then evaporated until dried. The residue was re-dissolved with 3 mL of deionized water/acetonitrile 80:20 *v*/*v*. 

For all types of the prepared samples, in the last step, the solutions were filtered through PTFE before being injected into UHPLC-ESI-MS/MS.

### 2.5. Condition and Instrument Optimization 

The separation was performed on a liquid chromatograph (Thermo Fisher Scientific, Burladingen, Germany). The UHPLC/ESI–MS system, a Thermo Fisher Scientific Ultimate 3000 (UHPLC), consists of an HPG-3200SD high-pressure gradient pump, a WPS-3000 RS thermostatted split-loop autosampler and a TCC–3000 RS column thermostat. The guard column, an Accucore RP-MS 10 × 2.1 mm 2.6 µm Defender Guard, was connected with RP-MS HPLC column C18 (dimensions: 100 × 2.1 mm and particle size: 2.6 µm) for separation of the chromatogram of steviol glycoside and its derivatives. The conditions were modified from those of Geuns et al. [43]. The column temperature was set at 30 °C, and the injection volume was 5 μL. Mobile phase A was 0.05% formic acid in acetonitrile and mobile phase B was 0.05% formic acid in deionized water. The flow rate of 0.4 mL min^−1^ was used for gradient elution. The gradient elution steps were performed within 13 min (Table 1).

The detection system consisted of a TSQ Quantis™ Triple Quadrupole mass spectrometer. Nitrogen was used as sheath gas (50 Arb), auxiliary gas (10 Arb) and sweep gas (1 Arb). The sample vaporization and ion transfer tube temperatures were set to 350 °C and 300 °C, respectively. Ionization was used with a heated ESI probe operating in negative ion mode and with the select reaction monitoring (SRM) mode. Regarding MS conditions, the determinations were carried out in negative ion mode as the analyte signal was much higher (about 10 times) than that in the positive ion mode [2]. Data acquisition and instrument control were managed with Chromeleon 7 by using the precursor ion and two productions of each compound along with the retention time to confirm a compound (Table 2).

### 2.6. Method Validation (MV) for Determination of Steviol Glycosides

In this study, MV was performed by following the ASEAN guideline [44] and the guideline for laboratory accreditation in Thailand, Department of Science Service, Ministry of Industry, which is based on “The Fitness for Purpose of Analytical Methods A Laboratory Guide to Method Validation and Related Topics” [30] for validation of analytical procedures. The following parameters were assessed: specificity, linearity, matrix effect, range, accuracy, precision, LOD and LOQ for MV.

#### 2.6.1. Specificity

In this study, the separated chromatogram of the sample was compared with the chromatogram of the standard by identifying whether the active pack presented at the same retention time. A blank and a mobile phase (eluent) were injected to study specificity, expecting that the active peak would not be observed at the retention time (RT) of the analyte. The relative percentage difference (%RPD) of RT was used for calculations, as is shown in the below equation,
(1)$$\%RPD=\frac{\left|{RT}_{1}-{RT}_{2}\right|}{\left(\frac{{RT}_{1}+{RT}_{2}}{2}\right)}\times 100$$
where RT_1_ is the high value of the mean retention time of the analyte and RT_2_ is the low value.

#### 2.6.2. Linearity of Analysis Results for Steviol Glycoside and Its Derivatives 

Linearity is a directly proportional relationship between the response and concentration of the analyte in the matrix over the range of analyte concentrations of interest. For each concentration, a percentage of relative standard deviation (%RSD) is considered with the criterion of about 15% [45]. In addition, a good linearity of the coefficient of determination (R^2^) > 0.99 is taken into consideration [46]. In this study, the series of mixed standard working solutions (0.2, 0.3, 0.4, 0.5, 0.8 and 1.0 mg L^−1^) was investigated in triplicate using UHPLC–ESI–MS/MS.

#### 2.6.3. Matrix Effect (ME)

The study of the ME was performed in 3 types of matrices: a non-alcoholic beverage, yogurt and snack. They were spiked with a stock mixture of steviol glycoside and its derivatives with three levels of concentration of 0.2, 0.5 and 1 mg mL^−1^. Each concentration was tested in seven replicates. There are many different methods for calculating the matrix effect. In this study, the ionization suppression or enhancement effect was calculated as the percentage of an absolute matrix effect (%ME_A_) by comparing the peak area of the standard with the peak area of the standard when added to a sample matrix. The equation for the calculation is shown below [12,47].
(2)$$\%MEA=\frac{Signal\;response\;of\;analyte\;in\;spiked\;matrix}{Signal\;response\;of\;analyte\;in\;solvent}\times 100$$
where the analytical signal of the spiked matrix is the peak area of the standard spiked with the analyte at the same concentration level into a sample blank, while the analytical signal of the standard in the solvent is the peak area of standard steviol glycoside in a solvent.

Normally, a %ME value of 100% indicates no effect on ion ionization. The coeluting compounds in a sample can cause a %ME lower than 100% or higher than 100% for ionization suppression and ionization enhancement, respectively [48,49,50]. According to the guidance on pesticide analytical methods for risk assessment and post-approval control and monitoring purposes [51,52], matrix effect values of 100 ± 20% or 80–120% are considered suitable, meaning that a foreign substance has little or no effect on the analysis of the substance of interest.

#### 2.6.4. Precision and Accuracy

##### Precision

For our single-laboratory analysis using the same equipment, the same staff and either the same day or between days, the percentage of relative standard deviation (%RSDr) and Horwitz ratio (HorRat) were used as performance parameters for the precision of repeatability and reproducibility, respectively. The %RSDr [30] can be calculated as per the below equation, and an accepted range of %RSDr, in accordance with the study concentrations, was set as less than 20% [46].
(3)$$\%RSDr\;or\;RSDr=\frac{SD*100}{\bar{x}}$$
where $\;\bar{x}$ is the average concentration of the chemical of interest and SD is the standard deviation of the concentration of the chemical of interest.

The HorRat_r_ equation is also displayed below [53]: (4)$${HorRat}_{r}=\frac{{RSD}_{r}}{PRSD}$$

The predicted relative standard deviation (PRSD) can be calculated as per the equation below:PRSD = 2^(1−0.5 log C)^(5)
where C is the concentration found or added, expressed as a mass fraction. 

In this study, the precision was assessed through the analysis of the spiked blank sample with the three levels of standard concentration of 0.2, 0.5 and 1.0 mg L^−1^. This study was also performed in three types of sample matrices (beverage, yogurt and snack). The accepted HorRat was less than 2.0 [28].

##### Accuracy

In this study, the recovery was used to study the accuracy of the method. The three concentrations of 0.2, 0.5 and 1.0 mg L^−1^ were added to the three types of blank samples (beverages, yogurt and snack). The recovery was calculated using the below equation [30]: (6)$$Recovery\;\left(\%\right)=\frac{{\bar{x}}^{\prime }-\bar{x}}{x_{spike}}\times 100\;$$
where ${\bar{x}}^{\prime }$ is the mean value of the spiked sample (mg L^−1^), $\bar{x}$ is the mean value of the blank sample (mg L^−1^) and $x_{spike}$ is the added concentration (mg L^−1^).

The accepted range of % recovery was 70–120%, and the range in level was 0.1–1.0 mg L^−1^ [54].

#### 2.6.5. Limit of Detection (LOD) and Limitation of Quantity (LOQ) 

In this study, analyses of the spiked sample at concentrations of 0.2, 0.5 and 1 mg L^−1^ were performed with seven replications (*n* = 7). According to the Eurachem guide, “The Fitness of Purpose of Analytical Methods”, the LOD was calculated by multiplying the standard deviation of the lowest concentration by 3. The LOQ was also calculated in the same manner, where the standard deviation of the lowest concentration was multiplied by 10, as per the below equations [30].
(7)$$\mathrm{LOD}=3\;*{S_{0}}^{\prime }$$


(8)
$$\mathrm{LOQ}=10\;*{S_{0}}^{\prime }$$


The standard deviation (S_0_**’**) for the calculation of LOD and LOQ was calculated using the below equation,
(9)$${S_{0}}^{\prime }=S_{0}/\sqrt{n}$$
where S_0_ is the standard deviation of replicate measurements of test samples with low concentrations of analyte, S_0_′ is the standard deviation used for calculating LOD and LOQ and *n* is the number of replicate observations averaged when reporting results, where each replicate is obtained following the entire measurement procedure.

### 2.7. Calculation Equation for Steviol Equivalents

Normally, steviol glycoside and its derivatives must be expressed as a “steviol equivalent” because the molecular weights of various glycosides are different, but they all have the steviol structure as their backbone. The factor for converting steviol glycoside and its derivatives to steviol equivalents is shown in a Appendix A. The equation for calculating the steviol equivalent levels of steviol glycoside and its derivatives is demonstrated below [55]:[SE] = CF × [SG](10)
where [SE] is the concentration as a steviol equivalent, [SG] is the concentration of individual steviol glycosides and CF is the conversion factor, as listed in Appendix A, for the corresponding steviol glycosides.

### 2.8. Analysis of Steviol Glycoside and Its Derivatives in Foods and Beverages

To test the performance of the method for validation, food and beverage samples where the contents of steviol glycosides were specified on the label, three different foods and beverages were purchased from retail markets located in Bangkok and nearby provinces such as Nakhon Pathom, Nonthaburi and Samut Sakorn. For each type of sample, the products were mixed into a pooled sample, kept at −20 °C and analyzed within one month. Each sample was analyzed in triplicate. The %recovery and %RSD were used as quality control for ensuring the test results. 

## 3. Results and Discussion

### 3.1. Optimization of Instrument Conditions and Preparation of Sample

At the beginning of the method’s development, 300 μL of an individual derivative of steviol glycoside (1 µg g^−1^) was injected with the flow rate of 10 μL min^−1^ into the instrument to optimize the mass condition. Data on the mass spectrum of steviol glycoside studied and reported by JECFA and Perera et al. were used as references for the optimization process, which is summarized in a Appendix A [56,57]. The negative ion mode was used for investigating the MS spectra of each derivative. To obtain maximum sensitivity for identification and detection, the collision energy (CE) and the RF lens voltage were optimized for all derivatives using direct infusion to acquire the richest relative abundance of precursor and product ions. The optimized MS/MS parameters, qualitative ion pairs and the quantitative ion pair for each analyte were used for method validation (Table 2). The external calibration method was used to quantify the level of each derivative. The ion pair with the highest relative intensity was selected for quantification purposes, while the second and the third ion pairs were used for confirmation. Many authors have reported that the interferences or matrix in a sample can cause ion suppression or enhancement in the quantitative analysis with electrospray ionization (ESI) or atmospheric pressure chemical ionization (APCI) [58,59,60]. In this study, the ionization source was ESI; in comparison, APCI has fewer matrix effects than ESI [61]. 

The separated chromatograms of nine derivatives of steviol glycosides obtained under optimized conditions (Table 2) are shown in Figure 2. According to the separated chromatograms, rebaudioside D, rebaudioside A, stevioside, rebaudioside F, rebaudioside C, dulcoside A, rubusoside, rebaudioside B and steviobioside were sequentially eluted at retention times of 4.139, 4.856, 4.895, 5.113, 5.214, 5.303, 5.693, 6.347 and 6.478 min, respectively. The developed analysis method in our study achieved the separation and detection of the nine steviol glycosides within 7 min, which is a very short separation time. The most desired steviol glycosides with the highest sweetening power (stevioside and rebaudioside A) and minor glycosides (dulcoside A, steviolbioside, rubusoside and rebaudioside B, C, D and F) can be separated with a high resolution based on mass identification using the validated method. As can be observed in the chromatograms, owing to the structural similarities of steviol glycosides, achieving baseline separation of steviol glycosides, particularly between stevioside and rebaudioside, is quite challenging. Attempts have been made in several previous studies, focusing on the UHPLC system equipped with MS. These studies have encountered many problems such as the chromatogram not showing, a long run time and a resolution reliant on the tandem mass detector [62,63].

Sample preparation and cleanup are the primary steps in most analytical methods that ensure the sample is suitable for the later analytical steps. From several research studies [42,64,65], we can surmise that the key points to consider for sample preparation are the different concentrations of intense sweeteners and the differences in chemical properties among these compounds (solubility and thermal stability) [66], affecting the use of a single extraction method to prepare different types of samples. In general, sample preparation methods including partition, filtration, centrifugation, sonication, protein precipitation, dilution and various forms of extraction are widely used [61]. In this study, the preparation of the beverage samples was easier than the other two samples of yogurt and a snack. 

For the beverage sample, the pretreatment processes were dissolving, vortexing, sonication, diluting and filtration. The pretreatment techniques used in this study were similar to those in several previous studies [43,64]. Uses of different types of solvents for dilution of samples were found. For example, drink samples and liquid sweeteners were diluted with H_2_O/ACN (80:20 *v*/*v*) to obtain analyte concentrations within the linear range of the calibration curve before being filtered through a PTFE syringe filter [43]. A few steps to prepare the sample were also found in the study by Romina Shah et al. [64]. For drink samples, dilution with H_2_O/ACN (80:20 *v*/*v*) and filtration through a PTFE syringe filter directly into HPLC autosampler vials were applied [67]. The pretreatment procedure could provide complete dissolution of the matrix, resulting in a transparent solution. 

Yogurt samples represent a much more complex mixture of ingredients than beverages; thus, appropriate sample preparation and clean-up prior to UHPLC-ESI-MS/MS analysis are required to ensure better long-term performance of the instrument and reduce ion suppression effects [68]. In this study, the yogurt sample was treated via dissolving, heat treatment, precipitation, dilution, centrifugation and filtration. Similar to previous studies, different temperatures were used in the extraction of SGs from this dairy product [19,69]. In this case, we applied temperatures of 60 °C and 75 °C for sample pretreatment, and a corresponding variation in the incubation times was used, with times being 1 h and 21 h, respectively [68,69]. Some studies used an acidic solution and extraction buffer, such as formic acid and N,N-diisopropylethylamine (DIPEA), for sample pretreatment [64], but this process was not applied in our study. We adopted the use of Carrez solutions to remove proteins and fat from dairy samples, which was found in several previous studies [12,19,70]. 

The snack sample consisted of complex ingredients. The steps in the sample preparation were similar to those for preparing the yogurt sample, with the only difference being the use of a solution for dissolving. A similar type of sample (fish cake) to that tested by Park et al. was used here to study the performance of Park et al.’s test method [71].

The following filtration processes were applied in our study: debris filtration through Whatman No. 1, SPE filtration for selective sample preparation and purification and hydrophilic PTFE membrane for purification prior to the chromatographic analysis and protection of expensive and sensitive analytical equipment. The use of SPE (Sep-Pak C18) in this study provided the percentage recovery of the main compounds (steviol and rebaudioside A) within the accepted criteria of 70–120%. There have been reports of using the Sep-Pak C18 cartridge to purify analytes from complex matrices with high-fat contents as intense sweeteners (including steviol glycosides) such as for chocolate and dairy products [70]. As is widely reported in the literature, SPE represents the most efficient way to overcome ME by purifying the sample and ridding it of interferences. The use of SPE must be focused on two steps: the first is choosing the sorbents that match the properties of the analytes of interest better [72,73], and the second is a proper selection of washing and elution solvents [74]. Within operational procedures, SPE may lessen matrix effects through the reduction or elimination of interferences [61].

Before injecting the treated sample into the LC system, the PTFE filter (0.22 µm, 13 mm) was used in this study, providing good percentage recoveries for all analytes of interest in all types of samples with the range of 80.12–118.38%. Similar to the present study, a good recovery rate of 98.4% to 100.5% was reported when using PTFE (0.45 µm, 25 mm) because the adsorption of target compounds through the PTFE filter was very low [75]. 

To introduce the samples into the LC system, in the present study, reduction in the viscosity of treated samples before injection was performed with dilution levels of 1:10–1:30. Some studies have shown that the presence of interfering compounds at a higher concentration can increase the viscosity and the surface tension of the droplets, which change the efficiency of their formation and evaporation. Changes in the liquid phase can alter the amounts of charged ions in the gas phase in MS-based analysis [61].

In this study, Accucore RP-MS HPLC column C18 (dimensions: 100 × 2.1 mm and particle size: 2.6 µm) was used for the separation of steviol glycoside and its derivatives. This type of column is classified in the modern sub-3 µm, solid-core particles (SCPs), with a porous shell (1.7 µm) surrounding a non-porous core (0.5 µm), ending up with a final particle size of 2.6 µm for packed particles [76], which has been shown to have high efficiencies and enable compatibility with both HPLC and UHPLC platforms [77]. There have been studies on the variability among three brands of columns packed with 2.6 μm or 2.7 μm SCPs [78], where the %RSD results of six analyses for the efficiencies of 100 × 2.1 mm columns were in the range of 4.6–6.8%. In the present study, the separation of a chromatogram of each steviol glycoside was achieved through the gradient elution conditions of the mobile phase, providing the nine chromatograms of steviol glycoside derivatives (Figure 2). The rapid analysis of the major steviol glycosides (three to five compounds of steviol glycoside derivatives) was developed using several UHPLC methods. For example, Wang et al. [79] compared different UPLC columns for the separation of five steviol glycoside derivatives. They reported that the best separation and peak shape was found for the UPLC HSS T3 column (dimensions: 100 × 2.1 mm and particle size: 1.8 µm). This type of column is classified in the sub-2 µm, totally porous, solid-core particles (SCPs) [80]. In a previous study similar to the present study, the solvent composition of acetonitrile with 0.05% formic acid (*v*/*v*) and 0.05% formic acid in water was used for the separation of five steviol glycoside derivatives, revealing that this set of mobile phases provided optimal chromatographic separation [65]. 

From all the discussions on method development and optimization, it is important to point out that there is no universal strategy, and in many cases, several approaches must be combined to achieve adequate quantitative results.

### 3.2. Method Validation

The validation of the UHPLC–ESI–MS/MS method for the quantification of the nine derivatives of steviol glycosides was performed by investigating the following quality parameters: selectivity, linearity, matrix effect, precision, recovery, LOD and LOQ in three types of matrices—beverage, yogurt and snack. The developed method was tested for its capacity to determine the presence and concentrations of these analytes in the sampled food and drink items. Gardner et al. [81] reported that drinks, candies and yogurts are often chosen for research in this field because sweeteners are widely used in these commonly consumed products. 

#### 3.2.1. Selectivity and Specificity

Selectivity allows an analyte of interest to be isolated from the mixture and gives confidence that the correct component is being measured. Good selectivity leads to reduced noise, allows detection at low levels and can avoid isobaric interferences and matrix effects [82]. In this study, the RT was the one parameter used to calculate the random error between two measurements. The RT of each steviol glycoside in spiked samples was compared with the mean RT of the corresponding standard. This study was performed in the three types of matrices already mentioned, with the concentration range of 0.2–1.0 mg L^−1^. The result showed percentage differences in RTs with ranges of 0.41–1.10%, 0.00–0.29% and 0.00–0.29% for the beverage, yogurt and snack, respectively (Table 3). All of the differences in RT were less than 5% according to the EU 2002 criteria [83]. 

Generally, retention time shifts are indicative of leaks, pump malfunctions, the type of column, column dimensions, degradation of the column, changes in column temperature or mobile phases [84]. In MS-based analysis, given the presence of high-resolution mass analyzers, some compounds have the same molecular formulae and similar tandem mass spectra and can be falsely positively identified; for instance, stevioside has the potential to interfere with the quantitation of Reb A [85]. The use of RT can improve the identification and quantification of the *m*/*z* ratio in LC–MS analysis. The mass analyzer, as a triple quadrupole system, consists of three quadrupoles; Q1 and Q3 work as mass filters, while Q2 acts as a collision cell, providing good selectivity by eliminating false positive peaks [86,87]. 

In this study, the problem of the RT shift was primarily attributed to column blockage caused by the complex nature of the sample used. The sample preparation techniques of filtration with a paper filter, SPE and PTFE were employed to mitigate column blockage. During the development of the analysis method, it was found that the filter membrane had a certain effect on the recovery rates of the analytes, so the selection of filter membrane was mainly optimized. This action has rarely been discussed in previous studies. From the results, we found that SPE (500 mg) and 0.2 µM PTFE could increase recovery rates to an acceptance criterion. These practices were in accordance with the study by Han et al. [75]. Zhou et al. [61] found that the target compounds filtered through PTFE had good recoveries in the range of 98.4% to 100.5%. 

#### 3.2.2. Linearity

Matrix-matched calibration curves, when using fortified food samples with no detectable sweeteners, were established to assess the linearity of the method. The coefficients of determination (R^2^) of each derivative of steviol glycoside in the beverage, yogurt and snack were in the ranges of 0.9911–0.9990, 0.9939–1.0000 and 0.9973–0.9999, respectively (Table 4). Calibration data of the method subject to validation showed that the detection response for all analytes of interest in all matrices could provide linearity (R^2^) > 0.99, indicating good linearity [88], meaning that the method could perform well in quantifying the amounts of nine steviol glycoside derivatives. The calibration data from the spiked samples and standard solutions were compared, and they indicated that analyte accuracy was within the experimental error limits. Therefore, calibration curves for standard solutions were used for the quantitation of the steviol glycosides in the samples. The relative standard deviation (%RSD) is the criterion used for checking the precision of the linearity in an analytic food matrix. According to the criteria of the AOAC, an accepted %RSD is less than 15% for the concentration range of 0.1–1.0 mg L^−1^ [46]. The results showed that the %RSD for the beverage, yogurt and snack were in the ranges of 0.25 to 14.22, 0.27 to 6.23 and 0.76 to 11.55, respectively.

#### 3.2.3. Matrix Effect (ME)

The analyzed samples not only contained the analyte of interest but also contained the matrix, which refers to the other components of the sample. The matrix in the sample can interfere with or affect the obtained analysis results, which is referred to as the “matrix effect” [71]. The matrix effect may have a significantly negative impact on LC–MS analysis, i.e., the ionization efficiency of the analyte, particularly when ESI is used, causes suppression or enhancement of the analyte signal, thus affecting quantification, possibly leading to incorrect results [89]. The extent of the matrix effect depends on the type of matrix itself and the variability between samples of the same type, preparation procedure and chromatographic and MS analysis conditions, as well as chemical properties of the analyte and the interactions between the analyte and the interfering coelution [90]. Matrix effect determination allows for the assessment of the reliability and selectivity of an existing analysis method, including the UHPLC–MS/MS method. 

In this study, the standard addition method was applied to study the ME. Then, the peak area of the standard in solution and standard spiking in the sample matrix (after applying the whole method) were calculated in percentages of the absolute matrix effect. The percentages of the absolute matrix effect were studied in the three types of samples. The three concentrations of 0.2, 0.5 and 1.0 mg L^−1^ were added to each type of sample. The study of matrix effects in each was conducted in triplicate and the results are presented in terms of the absolute matrix effect (ME_A_), which corresponds to the signal suppression or enhancement [91]. The percentages of the absolute matrix effect (%ME_A_) of the samples’ matrices were within the accepted criteria of 80–120% (Table 5), meaning that the foreign substances did not affect to the analysis of the nine derivatives of steviol glycoside. 

The %ME_A_ results for the studied concentrations of steviol glycoside and its derivatives were in the ranges of 88–117%, 73–98% and 71–125% for the beverage, yogurt and snack, respectively. As mentioned, the acceptance criterion is that the percentage of %ME_A_ must be within the range of 80–120%. For ion suppression, a low % ME_A_ was found in the yogurt matrix. For analysis of rebaudioside B, % ME_A_ results of 76.14 and 79.43% were found at 0.2 and 0.5 mg L^−1^, respectively. For steviolbioside, %ME_A_ results of 73.29 and 75.29% were found at 0.2 and 0.5 mg L^−1^, respectively. Ion enhancement was found in the snack matrix with %ME_A_ results of 125.14 and 121.43% for analyses of rebaudioside A and dulcoside A at 0.2 mg L^−1^, respectively. As mentioned above, the ME can cause either suppression or enhancement of the compound signal, but the former is more common [92], as in the present study, where we found that ion suppression was greater than ion enhancement. However, almost all %ME_A_ results were in the range of 80–120%, indicating that the matrix had no effect on the analysis of steviol glycoside and its derivatives when we used the selected analysis method [52].

#### 3.2.4. Precision and Accuracy

Precision is the parameter used to determine the closeness of a set of individual analyses. The %RSD, for within-day results and HorRat, represented reproducibility and was used for estimating the precision of the analysis method. In principle, the study of precision must be taken at low, medium and high levels of concentration of analyte, and a minimum of six determinations for each concentration is recommended. In this study, the three levels of standard steviol glycoside and its derivatives, 0.2, 0.5 and 1.0 mg L^−1^, were spiked in three types of sample matrices (beverage, yogurt and snack), and each level was performed for seven replications. For the beverage, the %RSD results for the concentrations of 0.2, 0.5 and 1.0 mg L^−1^ were in the ranges of 7.39–9.30%, 2.51–4.27% and 4.28–7.60%, respectively. For the yogurt, the %RSD results for the concentrations of 0.2, 0.5 and 1.0 mg L^−1^ were in the ranges of 1.10–7.52%, 2.34–4.55% and 1.79–5.69%, respectively. For the snack, the %RSD results of 1.25–3.08%, 1.28–3.73% and 1.64–5.39% were found at the concentrations of 0.2, 0.5 and 1.0 mg L^−1^, respectively (Table 6). As these results show, the %RSD of all food matrices was within an accepted range of less than 20% [46]. Table 7 shows the results for the HorRat values of all derivatives of steviol glycosides in each type of sample matrix. The obtained HorRat results range from 0.05 to 0.48; since they are less than 2, they are, therefore, within the performance criteria of the Commission Regulation (EC) No. 401/2006 [54].

For the beverage, the average % recovery results of all derivatives were in the ranges of 74.93–100.53%, 102.12–114.67% and 94.50–108.06% for the concentration levels of 0.2, 0.5 and 1.0 mg L^−1^, respectively. For the yogurt, the average % recovery results of all derivatives were in the ranges of 70.00–89.42%, 75.36–94.04% and 81.18–97.93% for the concentration levels of 0.2, 0.5 and 1.0 mg L^−1^, respectively. For the snack, the average % recovery results of all derivatives of the concentrations of 0.2, 0.5 and 1.0 mg L^−1^ were in the ranges of 84.49–119.14%, 73.92–115.45% and 72.43–108.85%, respectively. The detailed % recovery results are depicted in Table 8. In all of the results, the percentages of recovery were within an accepted range of 70–120% [28,55].

#### 3.2.5. Limit of Detection (LOD) and Limit of Quantitation (LOQ)

The results of LOD and LOQ are shown in Table 9, which reveals that the LOD and LOQ of steviol equivalents for the beverage product were in the ranges of 0.050–0.078 and 0.165–0.261 mg kg^−1^, respectively. For the yogurt, the LOD and LOQ of steviol equivalents were in the ranges of 0.003–0.028 and 0.011–0.093 mg kg^−1^, respectively. For the snack, the LOD and LOQ of steviol equivalents were in the ranges of 0.013–0.029 and 0.045–0.098 mg kg^−1^, respectively. 

According to the principles and methods for the risk assessment of chemicals in food [93], when conducting a chemical risk assessment, it is important to have detailed information on the data source, survey type or design, sampling procedures, sample preparation, analytical method, LOD or LOQ and quality assurance procedures. In addition, EFSA [38] also recommends ways to deal with “not detected” analysis results or left-censored data in estimating dietary exposure to the chemical. Generally, the risk assessor should assume that chemicals may be present in the food or drink sample at undetectable levels, and they should prescribe at which values it is appropriate to consume a substance as food or drink, i.e., those lower than the LOD or LOQ. Commonly, there are three scenarios of substitution methods used for dealing with “not detected” analysis results: (1) lower bound, where if the analysis result is lower than the LOD and LOQ, the result is substituted with “0”; (2) middle bound, where if the analysis result is lower than the LOD and LOQ, the result is substituted with 1/2 LOD or 1/2 LOQ, respectively; and (3) upper bound, where if the analysis result is lower than LOD or LOQ, the result is substituted with LOD or LOQ, respectively. Practically, the lower bound and the upper bound are used for chemicals mostly found in food, such as natural contaminants, nutrients and toxins from pathogens. Both low and high values should be taken to calculate the average value of food chemicals, which is then further used for calculating dietary exposure to chemicals. The use of the middle bound to calculate the mean value is limited, but if more than 50% of the analysis samples are lower than the LOD or LOQ, then using the median obtained from the dataset is better than the average value. In addition, there are suggestions that we should consider the “not detected” value in various cases, such as from a team of GEMS/food project experts [94].

### 3.3. The Analysis of Steviol Glycoside and Its Derivatives in Foods and Beverages

The method subject to validation, UHPLC-ESI-MS/MS, was used to quantify the concentrations of steviol glycoside and its derivatives in 38 samples of foods and beverages. The concentration of each derivative was presented in steviol equivalents with the unit of mg kg^−1^ (Table 10). All types of samples were analyzed in triplicate. Precision and accuracy (%Recovery) were studied as the quality control of the analysis. The results showed that all types of samples contained rebuadioside A (Reb A). The second most abundant compound found in the analyzed samples was stevioside (SV), followed by rebuadioside B (Reb B). This indicates that Reb A and SV are used in high amounts in commercial products because of their high levels of sweetness. The precision of the analysis (%RSD) was in the range of 0.5–13.97%, which is within the accepted criterion of 15%. For % recovery, the accepted range is 70–120%. The results showed that almost all analyses of steviol glycoside and its derivatives were in the accepted range. The lowest % recovery of 65.83% was found in the analysis of rebuadioside B and the highest % recovery of 123.62% was found in the analysis of dulcoside. According to the Notification of the Ministry of Public Health No. 418 B.E. 2563 (2020), steviol glycosides are permitted for use as INS: 960a and INS: 960b(i) with varying amounts as follows. Beverages are classified in group 14.0, which allows maximum use levels of steviol glycosides in the ranges of 40–350 mg kg^−1^. Yogurt is classified in group 0.1 of dairy products and analogs, which allows maximum use levels of steviol glycosides in the ranges of 70–330 mg kg^−1^. Snacks are classified in group 15.0 of ready-to-eat savories, which allows maximum use levels of steviol glycosides in the ranges of 40–350 mg kg^−1^. The average levels of each steviol glycoside were within the levels allowed under Thai regulation. In addition, the average levels of steviol glycoside equivalence were within the Codex Alimentarius standard of less than 330, 200 and 170 mg kg^−1^ for dairy-based desserts, beverages and snacks, respectively [95]. Additionally, these levels conform to the JECFA standard for beverages, desserts and yogurt, which is less than 500 mg kg^−1^ [96].

In the overall context of this study, MV provides the most significant advantage in establishing a high degree of confidence for both developers and users. Although validation activities may seem costly and time-consuming, they ultimately eliminate frustrating repetition and improve time management. Through MV activities, valuable insights can be gained from related published works, saving time and reducing costs associated with the procurement of various chemicals and assembly parts, as well as enhancing proficiency with advanced instruments. Consequently, MV is essential when introducing a new method, when revising established methods or when these methods are employed in different laboratories by different analysts.

## 4. Conclusions

The suitability of the analysis across a wide range of sample types was taken into consideration during our UHPLC-ESI-MS/MS optimization and method validation. As a result, the developed method allowed for the separation and quantification of nine derivatives of steviol glycosides—rebaudioside D, rebaudioside A, stevioside, rebaudioside F, rebaudioside C, dulcoside A, rubusoside, rebaudioside B and steviobioside—within a very short separation time of approximately 7 min.

Full validation experiments were conducted in the laboratory under optimized conditions for sample preparation and UHPLC-ESI-MS/MS determination by spiking three levels of a standard mixture into blank samples. A major concern in quantitative LC–MS/MS analysis is the matrix effect, which can affect the accuracy, precision and sensitivity of a method. Accordingly, the matrix effect was evaluated during the method development to ensure we were producing reliable analytical data. All validated results were within the accepted criteria.

## Figures and Tables

**Figure 1 foods-12-03941-f001:**
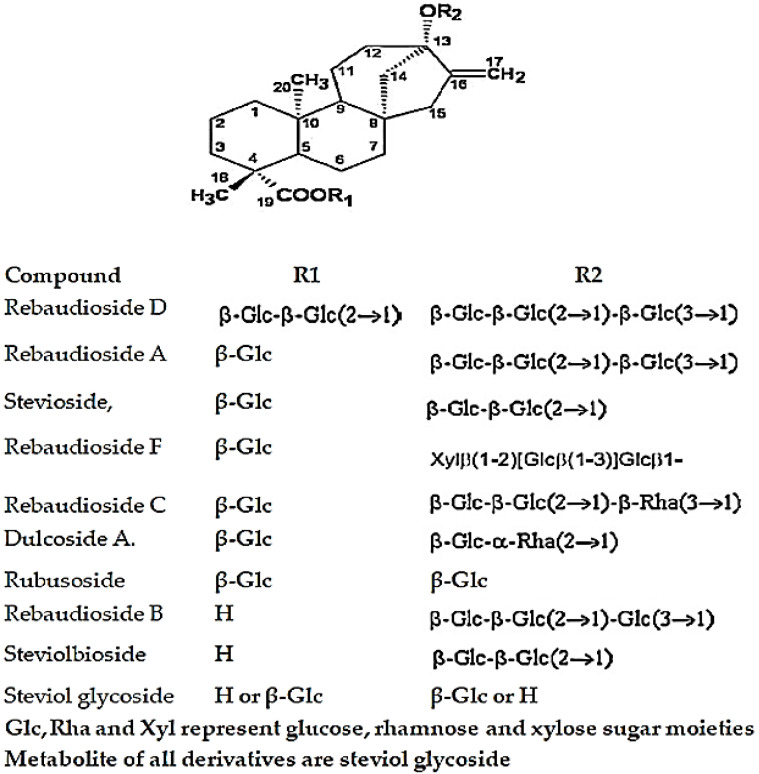
Structure of a steviol glycoside and its derivatives.

**Figure 2 foods-12-03941-f002:**
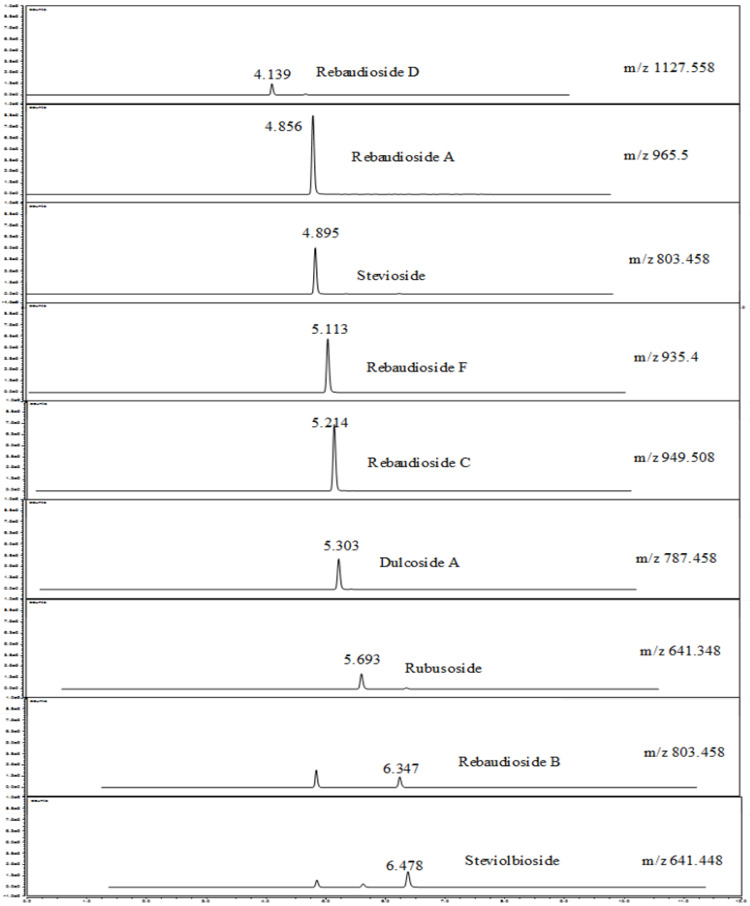
Chromatographic profiles of steviol glycoside standards: rebaudioside D, rebaudioside A, stevioside, rebaudioside F, rebaudioside C, dulcoside A, rubusoside, rebaudioside B and steviobioside.

**Table 1 foods-12-03941-t001:** Gradient conditions used for the UHPLC–MS/MS method.

Time (min)	% Mobile Phase A	% Mobile Phase B
0–1.9	10	90
2.0–4.9	30	70
5.0–6.9	35	65
7.0–8.9	90	10
9.0–12.5	10	90

**Table 2 foods-12-03941-t002:** Optimization of UHPLC–MS/MS conditions.

Sweeteners	Formula	RT * (min)	Compound					
			Precursor ion (*m*/*z*)	Product Ion				S-lens (RF Voltage)
				Confirmatory Product Ions (*m*/*z*)	Collision Energy (eV)	Additional Product Ions (*m*/*z*)	Collision Energy (eV)	
Rebaudioside A	C_44_H_70_O_23_	4.856	965.5	803.45	27.87	641.36 317.29	55 55	280
Rebaudioside B	C_38_H_60_O_18_	6.347	803.458	641.29	47.42	413.08 317.17	54.07 53.65	280
Rebaudioside C	C_44_H_70_O_22_	5.214	949.508	787.38	32.2	641.29 479.24	55 55	280
Rebaudioside D	C_50_H_80_O_28_	4.139	1127.558	803.35	51.93	641.32 623.30	55 55	280
Rebaudioside F	C_43_H_68_O_22_	5.113	935.4	773.35	26.9	611.292 641.321	55 55	280
Rubusoside	C_32_H_50_O_13_	5.693	641.348	478.92	14.14	521.45 317.15	23.07 46.95	226
Dulcoside A	C_38_H_60_O_17_	5.303	787.458	625.45	18.1	479.29 317.25	55 55	272
Stevioside	C_38_H_60_O_18_	4.895	803.458	641.39	19.66	479.22 317.34	55 55	257
Steviolbioside	C_38_H_50_O_13_	6.478	641.448	479.17	42.32	461.29 317.37	48.39 42.49	280

* RT, retention.

**Table 3 foods-12-03941-t003:** The relative percentage difference (RPD) in RT for each derivative of steviol glycoside within each food group.

Analyte	RPD (%)		
	Beverage	Yogurt	Snack
Reb A	0.41	0.00	0.24
Reb B	1.10	0.18	0.18
Reb C	0.57	0.22	0.22
Reb D	0.41	0.29	0.29
Reb F	0.78	0.23	0.00
Rub	0.70	0.20	0.20
Dul	0.75	0.22	0.22
Stevio	0.41	0.24	0.24
Stebio	0.46	0.17	0.17

Reb: rebaudioside; Rub: rubusoside; Dul: dulcoside A; Stevio: stevioside; Stebio: steviolbioside.

**Table 4 foods-12-03941-t004:** Linearity of steviol glycoside and its derivatives in three types of samples.

Analyte	Beverage		Yogurt		Snack	
	R^2^	Linear Regression	R^2^	Linear Regression	R^2^	Linear Regression
Reb A	0.9954	y = 79,846x + 3076.40	0.9993	y = 8 × 10^6^x + 580,746	0.9995	y = 8 × 10^6^x − 253,645
Reb B	0.9911	y = 23,115x + 1642.30	0.9991	y = 2 × 10^6^x + 22,497	0.9993	y = 2 × 10^6^x − 71,164
Reb C	0.9990	y = 79,357x + 4082.30	0.9939	y = 1 × 10^7^x + 624,742	0.9988	y = 1 × 10^7^x − 555,332
Reb D	0.9931	y = 26,518x + 1389.70	0.9994	y = 4 × 10^6^x + 129,063	0.9962	y = 4 × 10^6^x − 123,139
Reb F	0.9942	y = 76,473x + 4149.00	0.9982	y = 8 × 10^6^x + 394,354	0.9990	y = 7 × 10^6^x − 308,079
Rub	0.9944	y = 29,488x + 1149.00	0.9992	y = 413,129x + 3182.2	0.9973	y = 450,167x − 17,830
Dul	0.9950	y = 22,672x + 2313.80	1.0000	y = 4 × 10^6^x + 95,758	0.9998	y = 5 × 10^6^x − 186,365
Stevio	0.9953	y = 48,117x + 4517.90	0.9990	y = 6 × 10^6^x + 305,800	0.9996	y = 6 × 10^6^x − 215,946
Stebio	0.9961	y = 32,307x + 3114.20	0.9990	y = 5 × 10^6^x + 111,242	0.9999	y = 7 × 10^6^x − 172,092

Reb: rebaudioside; Rub: rubusoside; Dul: dulcoside A; Stevio: stevioside; Stebio: steviolbioside.

**Table 5 foods-12-03941-t005:** Study of the matrix effect in three types of samples.

Analyte	Percentage Absolute Matrix Effect (Mean ± SD)								
	Beverage			Yogurt			Snack		
	0.2 mg L^−1^	0.5 mg L^−1^	1.0 mg L^−1^	0.2 mg L^−1^	0.5 mg L^−1^	1.0 mg L^−1^	0.2 mg L^−1^	0.5 mg L^−1^	1.0 mg L^−1^
Reb A	94.43 ± 6.55	117.43 ± 4.65	103.00 ± 4.28	88.29 ± 0.76	92.29 ± 1.80	96.86 ± 1.77	**125.14** ± 2.27	101.14 ± 2.67	100.43 ± 3.10
Reb B	89.14 ± 4.63	111.14 ± 3.18	101.14 ± 5.98	**76.14** ± 2.61	79.43 ± 3.51	84.43 ± 2.70	92.71 ± 3.09	80.14 ± 1.86	79.00 ± 1.41
Reb C	98.86 ± 5.87	110.00 ± 3.06	101.86 ± 4.38	88.14 ± 3.72	89.43 ± 1.81	97.71 ± 4.07	114.71 ± 2.56	103.43 ± 2.64	98.43 ± 5.32
Reb D	96.71 ± 6.73	108.57 ± 2.23	105.43 ± 5.56	86.29 ± 3.95	92.29 ± 2.21	94.57 ± 3.87	108.29 ± 6.13	111.57 ± 4.65	106.29 ± 2.56
Reb F	98.00 ± 6.06	110.71 ± 3.99	100.71 ± 6.02	84.29 ± 4.79	93.14 ± 3.08	92.43 ± 5.03	109.71 ± 2.98	95.43 ± 3.10	94.14 ± 1.57
Rub	95.43 ± 4.20	110.86 ± 2.97	104.71 ± 5.22	87.71 ± 4.75	94.14 ± 3.39	94.57 ± 3.21	108.29 ± 3.40	92.43 ± 3.15	95.43 ± 2.07
Dul	97.86 ± 4.74	110.86 ± 2.73	93.71 ± 6.40	87.57 ± 3.51	94.86 ± 2.91	97.86 ± 4.60	**121.43** ± 1.81	101.43 ± 3.31	99.71 ± 2.36
Stebio	87.57 ± 5.09	103.57 ± 3.21	96.71 ± 4.96	**73.29** ± 3.25	75.29 ± 3.25	82.57 ± 2.57	82.43 ± 3.10	73.00 ± 1.53	71.43 ± 3.10
Stevio	93.14 ± 4.67	106.29 ± 3.59	97.86 ± 4.74	92.57 ± 4.20	90.14 ± 3.67	95.71 ± 4.92	120.00 ± 1.83	96.86 ± 1.77	96.57 ± 1.62

Reb: rebaudioside; Rub: rubusoside; Dul: dulcoside A; Stevio: stevioside; Stebio: steviolbioside. The bold represents a %ME lower or higher than the guideline criteria of 80–120%.

**Table 6 foods-12-03941-t006:** The percentage of relative standard deviation (%RSD) for the analysis of steviol glycosides in a beverage, yogurt and snack at spiked levels of 0.2, 0.5 and 1.0 mg L^−1^.

Analyte	%RSD (%)								
	Beverage			Yogurt			Snack		
	0.2 mg L^−1^	0.5 mg L^−1^	1.0 mg L^−1^	0.2 mg L^−1^	0.5 mg L^−1^	1.0 mg L^−1^	0.2 mg L^−1^	0.5 mg L^−1^	1.0 mg L^−1^
Reb A	8.10	4.27	4.43	1.10	2.34	1.79	1.67	2.74	2.45
Reb B	7.39	3.39	6.30	3.59	4.55	3.49	2.84	1.96	1.77
Reb C	7.53	3.03	4.28	5.30	2.56	4.20	2.01	2.36	5.39
Reb D	8.95	2.51	5.52	5.69	2.42	4.30	2.60	3.73	2.15
Reb F	8.28	3.82	6.07	7.52	3.49	5.69	2.39	3.25	1.64
Rub	5.31	2.87	5.29	5.61	3.82	3.28	2.52	3.03	2.15
Dul	7.71	2.96	7.60	4.66	3.02	4.72	1.25	3.36	2.39
Stevio	7.80	3.91	5.35	5.75	4.29	5.13	1.41	1.28	1.69
Stebio	9.30	3.76	5.55	5.07	4.45	2.91	3.08	1.64	4.19

Reb: rebaudioside; Rub: rubusoside; Dul: dulcoside A; Stevio: stevioside; Stebio: steviolbioside.

**Table 7 foods-12-03941-t007:** HorRat values of the replication analysis of steviol glycoside and its derivatives in three types of samples.

Analyte	Precision (HorRat)								
	Beverage			Yogurt			Snack		
	0.2 mg L^−1^	0.5 mg L^−1^	1.0 mg L^−1^	0.2 mg L^−1^	0.5 mg L^−1^	1.0 mg L^−1^	0.2 mg L^−1^	0.5 mg L^−1^	1.0 mg L^−1^
Reb A	0.36	0.19	0.28	0.05	0.10	0.11	0.07	0.12	0.15
Reb B	0.33	0.15	0.39	0.16	0.20	0.22	0.13	0.09	0.11
Reb C	0.33	0.13	0.27	0.23	0.11	0.26	0.09	0.10	0.34
Reb D	0.40	0.11	0.34	0.25	0.11	0.27	0.12	0.16	0.13
Reb F	0.37	0.17	0.38	0.33	0.15	0.36	0.11	0.14	0.10
Rub	0.23	0.13	0.33	0.25	0.17	0.21	0.11	0.13	0.13
Dul	0.34	0.13	0.47	0.21	0.13	0.30	0.06	0.15	0.15
Stevio	0.34	0.17	0.33	0.25	0.19	0.32	0.06	0.06	0.11
Stebio	0.41	0.17	0.35	0.22	0.20	0.18	0.14	0.07	0.26

Reb: rebaudioside; Rub: rubusoside; Dul: dulcoside A; Stevio: stevioside; Stebio: steviolbioside.

**Table 8 foods-12-03941-t008:** Percentages of recovery of three levels of steviol glycoside and its derivatives in three types of samples.

Analyte	Mean of % Recovery (%)								
	Beverage			Yogurt			Snack		
	0.2 mg L^−1^	0.5 mg L^−1^	1.0 mg L^−1^	0.2 mg L^−1^	0.5 mg L^−1^	1.0 mg L^−1^	0.2 mg L^−1^	0.5 mg L^−1^	1.0 mg L^−1^
Reb A	100.53	114.67	104.53	82.71	92.72	97.25	119.14	103.33	100.04
Reb B	79.19	103.03	100.84	75.73	78.70	83.42	94.65	80.97	80.52
Reb C	93.75	111.60	108.06	79.89	89.97	94.60	116.15	100.39	99.71
Reb D	87.90	107.39	103.36	82.85	93.60	94.82	118.09	115.45	108.85
Reb F	88.50	108.82	98.68	77.34	92.93	92.31	113.84	94.07	94.20
Rub	96.15	112.10	102.78	87.77	92.49	93.68	103.58	92.69	93.72
Dul	86.24	110.49	94.57	87.39	94.04	97.93	118.38	100.84	100.04
Stevio	80.12	103.92	97.59	89.42	91.76	95.83	116.82	97.88	97.01
Stebio	74.93	102.12	94.50	70.00	75.36	81.18	84.49	73.92	72.43

Reb: rebaudioside; Rub: rubusoside; Dul: dulcoside A; Stevio: stevioside; Stebio: steviolbioside.

**Table 9 foods-12-03941-t009:** LOD and LOQ of each steviol glycoside derivative in three types of samples reported as steviol equivalents.

Analyte	LOD (mg kg^−1^)			LOQ (mg kg^−1^)		
	Beverage	Yogurt	Snack	Beverage	Yogurt	Snack
Rebaudioside A	0.060	0.003	0.015	0.201	0.011	0.049
Rebaudioside B	0.053	0.012	0.024	0.175	0.041	0.081
Rebaudioside C	0.052	0.016	0.017	0.175	0.053	0.058
Rebaudioside D	0.050	0.015	0.019	0.165	0.050	0.065
Rebaudioside F	0.056	0.022	0.021	0.187	0.075	0.069
Rubusoside	0.057	0.028	0.029	0.191	0.093	0.098
Dulcoside A	0.060	0.018	0.013	0.199	0.061	0.045
Stevioside	0.056	0.023	0.015	0.187	0.078	0.049
Steviolbioside	0.078	0.020	0.029	0.261	0.066	0.098

**Table 10 foods-12-03941-t010:** (1) Steviol equivalent contents (mg kg^−1^) in beverages. (2) Steviol equivalent contents (mg kg^−1^) in yogurt products. (3) Steviol equivalent contents (mg kg^−1^) in snack products.

**(1)**										
**Sample**	**Analysis**	**RebA** **(*n* = 20)**	**RebB** **(*n* = 3)**	**RebC** **(*n* = 2)**	**RebD** **(*n* = 0)**	**RebF** **(*n* = 0)**	**Rub** **(*n* = 0)**	**Dul** **(*n* = 0)**	**Stevio** **(*n* = 5)**	**Stebio** **(*n* = 0)**
Beverage (*n* = 20)	Mean ± SD (mg kg^−1^) (Min-Max)	84.60 ± 208.52 (0.96–886.16)	6.49 ± 6.07 (2.63–13.48)	2.08 (1.83–2.32)	ND	ND	ND	ND	4.03 ± 4.27 (1.32–11.52)	ND
	Precision (%RSD)	0.93–10.41	1.43–3.41	-	-	-	-	-	0.64–4.10	-
	Accuracy (%Recovery)	79.39–113.45	65.83–105.21	70.89–102.00	70.36–114.52	84.83–105.23	83.47–106.38	77.17–123.62	80.00–96.78	73.33–104.95
**(2)**										
**Sample**	**Analysis**	**RebA** **(*n* = 10)**	**RebB** **(*n* = 1)**	**RebC** **(*n* = 0)**	**RebD** **(*n* = 0)**	**RebF** **(*n* = 0)**	**Rub** **(*n* = 0)**	**Dul** **(*n* = 0)**	**Stevio** **(*n* = 7)**	**Stebio** **(*n* = 0)**
Yogurt (*n* = 10)	Mean ± SD (mg kg^−1^) (Min-Max)	13.00 ± 6.63 (2.02–23.85)	0.42 (0.42)	ND	ND	ND	ND	ND	5.33 ± 4.86 (1.19–11.17)	ND
	Precision (%RSD)	0.85–13.55	-	-	-	-	-	-	0.67–13.15	-
	Accuracy (%Recovery)	100.85–113.45	65.83–88.37	86.33–97.28	100.83–106.22	84.83–105.23	83.47–96.13	77.17–101.52	80.00–90.35	70.00–90.35
**(3)**										
**Sample**	**Analysis**	**RebA** **(*n* = 8)**	**RebB** **(*n* = 5)**	**RebC** **(*n* = 3)**	**RebD** **(*n* = 4)**	**RebF** **(*n* = 3)**	**Rub** **(*n* = 0)**	**Dul** **(*n* = 3)**	**Stevio** **(*n* = 5)**	**Stebio** **(*n* = 2)**
Snack (*n* = 8)	Mean ± SD (mg kg^−1^) (Min-Max)	67.49 ± 72.48 (1.18–168.67)	5.58 ± 4.63 (1.30–12.48)	19.69 ± 20.90 (7.30–43.83)	1.49 ± 1.70 (0.62–4.04)	4.67 ± 4.88 (1.81–10.30)	ND	18.40 ± 19.12 (6.85–40.48)	54.11 ± 91.73 (1.02–215.59)	8.80 (1.64–15.95)
	Precision (%RSD)	0.50–12.45	2.91–9.63	4.04–6.62	2.85–13.97	4.18–9.53	-	4.57–10.59	0.76–8.08	-
	Accuracy (%Recovery)	80.85–114.09	89.56–113.75	115.04–120.00	115.12–117.74	104.90–108.89	106.81–115.71	107.57–109.60	89.86–103.23	75.34–93.32

Reb: rebaudioside; Rub: rubusoside; Dul: dulcoside A; Stevio: stevioside; Stebio: steviolbioside.

## Data Availability

The datasets generated to obtain the results presented in this article are available from the corresponding author upon reasonable request (pharrunrat.tan@mahidol.ac.th).

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
