# Peer review of "Validation of UHPLC-ESI-MS/MS Method for Determining Steviol Glycoside and Its Derivatives in Foods and Beverages"

_foods, 2023, doi:10.3390/foods12213941_

Round 1

Reviewer 1 Report

Comments and Suggestions for Authors

The experimental design was reasonable, the experimental amount was sufficient, and the determination method of stevia side and its derivatives in food was comprehensively evaluated.

1. “risk assessment” used in the title was not accompany to the content of the manuscript, this paper was method validation. For risk assessment, the concentration of steviol glycosides, daily intake dose, toxicological evaluation et.al., should be considered. So, I suggest the author to modified the title.

2. Three kinds of samples were used for detection in this design, whether non-alcoholic beverage, yogurt, and snake can be categorized into foods and beverage as in the title?

3. Abstract: please add the range of precision.

4. There were about 13 kinds of glycosidic compounds (stevioside, rebaudioside A, rebaudioside B, rebaudioside C, rebaudioside D, rebaudioside E, rebaudioside F, rebaudioside M, rebaudioside N, rebaudioside O, dulcoside A, rubusoside, steviolbioside),However, there were 9 kinds of compounds reflected in Table 1, why only these 9 kinds of compounds were made?

5. Please normalize the format of Figure 1, and the numbers on the abscissa overlap.

6. It is recommended to supplement the standard curves of 9 steviol glycosides and their derivatives.

7. What is the selection basis for the concentration of steviol glycosides and their derivatives in the three types of samples (0.2/0.5/1.0 mg/L), please supplement it in your article.

8. Please invite one professional scholar to revise the English language of your manuscript.

9. In section 2.6.3, the title was “matrix effect”, but the validate method described in this section was recovery rate, please revise to make the manuscript accurately.

10. Three snakes were mixed when sample preparation, are the matrix of these three snakes the same?

11. Why mix three products as one sample, how about measured individually?

12. Line 626, the subtitle number is not in right order, please revise.

13. The instrument in line 450 was LC-QqQ-MS, but in other part of the manuscript, it was described as UHPLC-ESI-MS/MS, please make sure which instrument was using.

14. Line 105, the “UHPLC-ESI-MS/MS” was first present, the full name should be given when the abbreviation first used in the text.

Comments on the Quality of English Language

Please invite one professional scholar to revise the English language of your manuscript.

Author Response

Dear Reviewer 1

Thank you very much for reviewing our manuscript.  Our answering and response to your recommendation are in the attached document.

Best Regards,

Pharrunrat Tanaviyutpakdee

pharrunrat.tan@mahidol.ac.th

Institute of Nutrition Mahidol University

Salaya Phuttamon thon

Nakhon Pathom

73170

Reviewer 2 Report

Comments and Suggestions for Authors

Analysis of sweeteners such as steviol glycoside and its derivatives is important, and the development of novel method is highly desired. This paper described a method using HPLC-MS/MS for the determining of 9 steviol glycosides in various foods and beverages. The results is interesting, however, there are some concerns about the method and presentations.

1)    The authors should provide figures for real sample analysis.

2)    Chromatograms in Figure 1 may combine to reduce size and only show the informative part. Chromatograms are too small.

3)    In my understanding, the S-len RF voltage only affect the ion transfer, why the authors change the S-len RF voltage during the separation? How about the MRM time table for different standards?

4)    For standard preparation, how the authors accurately weight 0.001 g (1 mg) standards? Commonly used balance may not able to give the desired accuracy.

5)    For the sample preparation, why the authors use methanal for SPE solvent, while the analytes are all high polarity compounds? The purpose of SPE is not clear.

6)    Standards seems not fully separated by HPLC in Table 1, and the authors claim they are well separated with good resolution, please clarity.

7)    Table 3 provided very little information about the developed method, the deviation in retention time may mainly attributed to the instrument and column used.

8)    The reported linear range in section 3.2.2 is from 0.1 to 1 mg/L is quite very limited.

9)    Units are not consistent in the whole manuscript and need revise.

10)  Table 10 is quite difficult to read and may consider revise.

Comments on the Quality of English Language

The quality of writing is fine; however, the quality of figures and tables need improved.

Author Response

Dear Reviewer 2

Thank you very much for reviewing manuscript. Our answering and response to your recommendation are in the attached document.

Best Regards,

Pharrunrat Tanaviyutpakdee

pharrunrat.tan@mahidol.ac.th

Institute of Nutrition Mahidol University

Salaya Phuttamonthon

Nakhon Pathom

73170

Reviewer 3 Report

Comments and Suggestions for Authors

Dear authors,

The manuscript entitled " Method Validation for determination of steviol glycoside and its derivatives by UHPLC-ESI-MS/MS in foods and beverages: risk assessment aspect" validate method for determination of nine steviol glycosides and its derivatives in food and beverage products, using ultra high liquid chromatography electrospray mass spectrometry (UHPLC ESI MS/MS). It presents scientific relevance for Food, Chemistry, Medicine and others area. Some authors have published articles related to the theme of the manuscript found in databases (Sciencedirect, Pubmed, MDPI, Web of science, etc). However, you need to change some details/information in the Title, Abstract, Introduction, Material and Methods, results and discussion, and conclusions.

1. Title: Adequate! There is no information in “Abstract” on "risk assessment aspect". I suggest revising or removing this expression from the “title” of the manuscript.

2. Abstract: Adequate! But:

- The abstract is well written, with details of the validated method. There is no information on "risk assessment aspect". I suggest revising or removing this expression from the “title” of the manuscript.

- To replace “mg/mL” and “mg mL-1”, and throughout the manuscript (including in tables).

- To replace “µg/g” and “µg g-1”, and throughout the manuscript (including in tables).

- At the end, I suggest highlighting the advantages of the study and methods.

- The keywords “Revalidation of method” and “Chemical risk assessment”. The words do not appear in the title or abstract! I suggest reviewing! Has the proposed method been validated before? Why “Revalidation of method”?

3. Introduction section:

- Page 2, line 68: To replace “mg/kg” and “mg Kg-1”, and throughout the manuscript (including in tables).

- I suggest at the end of the introduction, to highlight the "innovative" proposal of the method, as well as the advantages / disadvantages.

4. Materials and methods section: The methodological proposal is appropriate to the manuscript, but I suggest:

- Page 3, line 118, in “2.2 Standard solution of steviol glycoside and its derivatives” section: To replace “mg/L” and “mg L-1”, and throughout the manuscript (including in tables).

- Page 3, in “2.3 Sample preparation and extraction” section: How were the samples acquired? How were the samples stored until the time of analysis? What is the storage time?

- Page 9, in “2.8 Analysis of steviol glycosides and its derivatives in foods and beverages” section: How were the samples acquired? How were the samples stored until the time of analysis? What is the storage time? I suggest inserting more information about the composition of the samples.

5. Results and discussion

- Page 9, line 306, in “3.1 Optimization of instrument condition and preparation of sample” section: - To replace “μL/min” and “μL min-1”, and throughout the manuscript (including in tables).

- Page 13, in “3.2.1 Selectivity and Specificity” section: Long paragraph! I suggest splitting it into 2 or 3 paragraphs.

- Pages 15-16, in “3.2.3 Matrix effect (ME)” section: Long paragraph! I suggest splitting it into 2 or 3 paragraphs. Idem for “3.2.5 Limit of detection (LOD) and Limit of quantitation (LOQ)” and “6. The analysis of steviol glycosides and its derivatives in foods and beverages” sections.

- There is no monitoring/assessment of risks associated with the products. I suggest removing this expression from the title of the manuscript.

- I suggest, at the end of the "results and discussion", to write a paragraph summarizing the findings and their impacts on the research proposal.

6. Conclusion: Adequate, but I suggest including disadvantages/limitations of the method and the study!

7. Table and Figures: Adequate. To review the concentration units!

8. References: Please, check if the references are in accordance with the journal's rules.

Comments on the Quality of English Language

The language (English) is satisfactory (but, I suggest the final revision)!

Author Response

Dear Reviewer 3

Thank you very much for reviewing manuscript. Our answering and response to your recommendation are in the attached document.

Best Regards,

Pharrunrat Tanaviyutpakdee

pharrunrat.tan@mahidol.ac.th

Institute of Nutrition Mahidol University

Salaya Phuttamonthon

Nakhon Pathom 73170

Reviewer 4 Report

Comments and Suggestions for Authors

Minor remarks

Please, provide a blank space between quantity and unit except in the case of percentage.

Avoid the use of the first-person plural and avoid the phrase “we do that”, etc. The third-person singular is only acceptable for the scientific paper.

The terms that are mentioned in the manuscript for the first time can be presented in the abbreviated form. Also, the already-defined abbreviations should be used in the text.

Instead of “ml”, use “mL”.

The Greek letters should be present in italics.

Also, all Latin terms should be presented in italics. Please, carefully check that in the references list.

All minor remarks are depicted in the manuscript.

Major remarks

English grammar should be rechecked and language should be improved.

There are a lot of references for a research article. In its present form, it looks like a review article. Avoid lumping the references. Each reference should be discussed separately.

I suggest depicting the structure of analyzed compounds. Maybe, to do that in the Introduction section.

It is desirable to represent the separation of compounds in the real samples. I suggest the representation per one chromatogram for each sample.

Comments on the Quality of English Language

English language should be improved. Some grammatical errors should be corrected.

Author Response

Dear Reviewer 4

Thank you very much for reviewing manuscript. Our answering and response to your recommendation are in the attached document.

Best Regards,

Pharrunrat Tanaviyutpoakdee

pharrunrat.tan@mahidol.ac.th

Institute of Nutrition Mahidol University

Salaya Phuttamonthon

Nakhon Pathom 73170

Round 2

Reviewer 2 Report

Comments and Suggestions for Authors

The authors have addressed the concerns of the reviewers, I think this manuscript can be accepted for publication with some minor revise. For this is a research paper, the description for method and the presentation for results should keep concise. Another minor concern is that there are too many references have been cited in the paper and some of them are not necessary. For example, the rationales to conduct LODs and LOQs experiments are common sense for researchers in the field of food analysis and thus these citing are redundant. 

Author Response

Dear Reviewer 2,

Thank you very much for your effort in reviewing this manuscript. We provided our responses in attached document.

BR,

Pharrunrat Tanaviyutpakdee

pharrunrat.tan@mahidol.ac.th

Institute of Nutrition Mahidol University

Salaya Phuttamonthon

Nakhon pathom 73170
